# Delivering clinical studies of exercise in the COVID-19 pandemic: challenges and adaptations using a feasibility trial of isometric exercise to treat hypertension as an exemplar

Chris Farmer,[1,2] Ellie Santer,[1] Alan West,[3] John Darby,[3] Melanie Rees-Roberts,[1] Timothy Doulton,[2] Douglas MacInnes 🆔,[4] Jamie O'Driscoll,[5] Rachel Borthwick,[1] Tracy Pellatt-Higgins,[1] Katerina Gousia,[1] Vanessa Short,[6] Pauline A Swift,[7] Jonathan Wiles 🆔 [5]

For numbered affiliations see end of article.

**Correspondence to**
Dr Jonathan Wiles;
jim.wiles@canterbury.ac.uk

## ABSTRACT

The COVID-19 pandemic has significantly impacted on the delivery of clinical trials in the UK, posing complicated organisational challenges and requiring adaptations, especially to exercise intervention studies based in the community. We aim to identify the challenges of public involvement, recruitment, consent, follow-up, intervention and the healthcare professional delivery aspects of a feasibility study of exercise in hypertensive primary care patients during the COVID-19 pandemic. While these challenges elicited many reactive changes which were specific to, and only relevant in the context of 'lockdown' requirements, some of the protocol developments that came about during this unprecedented period have great potential to inform more permanent practices for carrying out this type of research. To this end, we detail the necessary adaptations to many elements of the feasibility study and critically reflect on our approach to redesigning and amending this ongoing project in order to maintain its viability to date. Some of the more major protocol adaptations, such as moving the study to remote means wherever possible, had further unforeseen and undesirable outcomes (eg, additional appointments) with regards to extra resources required to deliver the study. However, other changes improved the efficiency of the study, such as the remote informed consent and the direct advertising with prescreening survey. The adaptations to the study have clear links to the UK Plan for the future of research delivery. It is intended that this specific documentation and critical evaluation will help those planning or delivering similar studies to do so in a more resource efficient and effective way. In conclusion, it is essential to reflect and respond with protocol changes in the current climate in order to deliver clinical research successfully, as in the case of this particular study.

## INTRODUCTION

The recent outbreak of COVID-19[1] and the international response to impose 'stay at home' orders resulted in most clinical trials

## STRENGTHS AND LIMITATIONS OF THIS STUDY

⇒ The protocol developments documented provide a useful resource to other researchers and research managers tasked with delivering physical activity/ applied research trials in a 'post covid' environment.

⇒ The structured approach to the redesign of this clinical trial clearly highlights the advantage of having integrated and comprehensive patient and public involvement.

⇒ Recommendations are being made based on the delivery of a small-scale feasibility study.

⇒ The adaptations and implications identified may not be generalisable to all types of study design.

being suspended to recruitment, with the exception of those directly related to the pandemic. In May 2020, the National Institute for Health and Care Research (NIHR) issued guidance for restarting research paused due to COVID-19 for the UK[2] based on key guiding principles: viability (scientific, clinical, financial or practical reasons), safety, capacity and prioritisation. While these are fundamental to appropriate conduct of clinical trials, it is evident that changes during and following the pandemic present significant organisational challenges.

To help plan and undertake clinical research in the current climate, a structured approach to the redesign of clinical trials is described by Karzai *et al*[3] who draw attention to eligibility criteria, correlative studies, telehealth and partnerships, with particular emphasis on logistics of clinical trials and suggest that embracing change is vital.

The Medidata group recently identified that data completeness and collection have been a

key problem in the pandemic[4] and to mitigate some of the new challenges facing researchers, many regulatory authorities acknowledged the need to allow adaptations to trial recruitment, consent and monitoring.[5]

Here, we critically reflect on our approach to redesigning/amending a feasibility study of the impact of isometric exercise (IE) on arterial hypertension in otherwise healthy adults. This trial involved identification of people with stage 1 hypertension,[6] not on antihypertensive medication and with no relevant comorbidity. Participants were randomised to a period of isometric/static exercise (IE) and standard care 'lifestyle' advice (SCA) or control (SCA alone). End points included deliverability in the NHS (particularly primary care), fidelity of the intervention and impact on blood pressure (BP).[7]

We estimate that workload to deliver this project increased by >50% with the advent of COVID-19, for example, the Study Steering Committee needed to meet 3 monthly versus 6 monthly. The physical exercise nature of the trial also brought specific challenges for governance, safety and conduct, including evaluation of participant eligibility and informed consent along with the prescription of IE originally designed to be face-to-face. The study includes physiological measures of fidelity, for example, BP and heart rate (HR) responses to exercise, and remote monitoring systems needed to be developed for reliable collection of these data. By nature, exercise interventions require ongoing participant motivation[8] and additional methods to support this remotely were required. Because of the reduction in routine and face-to-face follow-up appointments, as well as changes to coding strategy in primary care, fewer patients were identified following searches of General Practitioner (GP) systems than in pilot work. Indeed, Dale *et al* suggested that nearly 500 000 fewer people were identified and treated for hypertension in mainland UK from March 2020 to 2021 compared with the previous year.[9] Paradoxically, it has been reported that the pandemic has heightened the need to focus on lowering the incidence of cardiovascular disease risk factors such as high BP and obesity.[10] While physical activity has been identified as a primary focus for cardiovascular disease prevention,[11] it is likely that pre-existing barriers to exercise prescription and promotion (eg, GP perceived status of exercise) have been exacerbated by the pandemic.[12] Recent research suggests that existing reticence among GPs based on lack of tradition, as well as lack of knowledge and validated tools,[13] is likely to have reduced the probability of exercise interventions being implemented. It was, therefore, necessary to reassess the capacity for NHS primary care staff to deliver the study and ultimately required a fundamental change to recruitment strategies. To help mitigate the impact COVID-19 has caused to research in the NHS, the Department for Health and Social Care (DHSC) launched their strategy regarding the future of UK research delivery,[14] recognising five key themes, three of which are directly addressed in the commentary

section of this paper: (2) patient centred, (3) streamlined, efficient and innovative and (4) research enabled by data and digital tools.

We aim to identify the challenges of patient and public involvement, recruitment, consent, follow-up, intervention aspects and primary care staff delivery of a feasibility study of exercise during the COVID-19 pandemic.

## COMMENTARY
### Patient and public involvement and engagement
Study delivery has benefited from lay members of the project management group, which allowed an integrated approach to redesign. Their previous experience and insight have been invaluable when commenting on important issues, offering a patient perspective to all elements of the redesign including: patient access to technology, use of personal protective equipment (PPE), optimising reminder texts (to mitigate attrition) and improving the participant documents and resources. As acknowledged by the NIHR,[15] patient and public involvement and engagement (PPIE) has been essential in successfully adapting the study for remote delivery during the pandemic and beyond.

The considerable time delays caused by COVID-19 restrictions along with the numerous amendments contributed to the significant increase in workload for the research team; arguably this disproportionately impacts on lay members whose continued involvement is no longer commensurate with initial commitment expectations. Interestingly, similar difficulties have resulted in many COVID-19 trials sacrificing valuable PPIE to meet time constraint pressures.[16] We have been extremely fortunate with the loyalty and commitment received from our public members and would advise anyone embarking on a funded research path to ensure they select these members with care. The importance of careful ongoing consideration of this aspect is reiterated in the UK-wide vision for the future of clinical research delivery which identifies the need to strengthen PPIE in research.[14]

### Changes to trial protocol and governance
As a result of COVID-19 restrictions, alterations had to be made to the study protocol, along with ethical amendments and this inevitably introduced significant delay to delivery of the study.[17] One major alteration involved moving all contact to remote means wherever possible, including the screening, baseline and follow-up visits. This clearly aligns with the DHSC's fourth key theme to ensure that research is enabled by data and digital tools.[14] This meant an additional remote appointment had to be added to screen and check patient eligibility, and clinic BP measurements were replaced by participant home BP readings using Omron M3 Intellisense machines, resulting in increased study costs. Also, this raised potential concerns regarding the accuracy of using this type of BP monitor:[18] despite the device being validated[19] and the

use of remotely observed BP measurements by a trained healthcare professional (HCP).

As a result of social distancing guidelines, participants were asked to carry out home BP readings with the investigator via video call. This was to ensure accurate home BP measurements according to National Institute for Health and Care Excellence guidelines.[6] The disadvantage of this approach was that participants needed access to technology which PPIE advised to avoid. Implementing this major change in delivery required additional equipment, such as webcam access, instructional resources (eg, videos) and alternative arrangements for those without IT access or ability, that is, free provision of smart technology or an additional visit. Thus, there were further logistical and cost implications associated with continued attempts to avoid inequity of access.

Due to reduced face-to-face contact with participants, it was necessary to develop a remote reminder system to mitigate increased risk of drop out. The sending of the messages was completely automated and made use of an SMS API provided by a large provider, with this system now reusable for future studies.

### Adaptation of participant identification searches and recruitment

The pandemic made recruitment more challenging for several reasons including a reduction of patients identified with stage 1 hypertension on GP records, reduced access to GP administrative staff and less provision of research active staff in primary care.

Searches of GP records yielded considerably fewer patients than pilot work had indicated. This is primarily attributed to: reduced attendances at GP clinics, suspension of routine health checks (eg, well man over 50), reduced recording of hypertension in primary care (suspension of some indicators in the quality and outcomes framework) and lack of repeat attendance for suspected hypertension. This hypothesis is supported by the findings of Dale et al,[10] who demonstrate a considerable reduction in numbers treated for incident hypertension during the pandemic. Anecdotal evidence also suggests that because GPs were concerned about being able to follow patients up (during the pandemic), many were commenced on antihypertensive medication immediately following diagnosis rather than allowing a period of lifestyle modification as per hypertension guidelines.[6] Since our study recruited untreated hypertensives, this rendered them ineligible. In future, initiatives such as rollout of the NHS community pharmacy BP check service[20] may mean potential participants for hypertension studies are identified outside the GP setting and supports the need for a more data-enabled research environment.[14]

As a result of persistent difficulties with recruiting in primary care, the study was approved for delivery in all NHS settings with additional direct to patient advertising. This targeted potential participants geographically via Facebook social media within reasonable travel distance

of a research site. In addition, those displaying interest in subjects that may predispose them to being attracted to the study were targeted. Users seeing the advert could click through to a prescreening survey to find out whether they were eligible to take part in the study and register their details.[21] This led to a greater number of potential participants (75% of those randomised) without involving any NHS staff time. This method also elicited a lower percentage of screen failures compared with GP screening and mail out (31% screen failure rate for direct advertising compared with 67% for mail out). Key learning has been the effectiveness of the prescreening survey in significantly reducing staff time (up to 12 hours of screening patient lists before mailout) and screen failure rates.

### Adaptations to consent

The requirement to reduce face-to-face contact with participants meant that, although consent remained a requirement, this process had to be managed virtually with the HCP on the video call and participants completing and signing an online form. In line with the drive for research to be enabled by digital tools,[14] this data are now captured straight into Qualtrics (online system), which is directly accessible by the research team. This allowed more efficient and accurate data handling without the need to transfer data from paper to database. In general, this worked well; however, there were some cases where it did not, primarily because patients were unable to access both Microsoft Teams and Qualtrics simultaneously.

### Changes to the intervention: prescription and development of IE training

The IE intervention used is a wall squat (figure 1) protocol, which involves leaning against a wall and squatting at an individual specific (knee joint) angle prescribed to elicit the required exercise intensity based on HR.[22]

To accurately prescribe an individual specific wall squat angle, participants must complete at least three-stages of a five-stage incremental IE test (IIET).[23] It was originally intended to subjectively preassess each patient's physical ability to meet this requirement during the

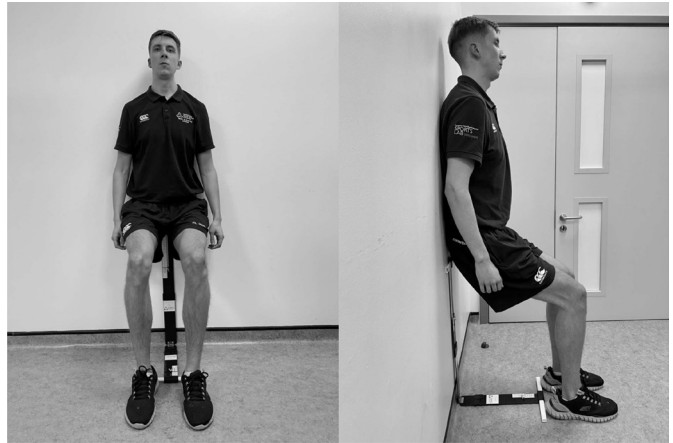

**Figure 1** Isometric wall squat exercise.

initial face-to-face screening visit. However, since this was replaced with a remote screening visit, it was not possible to complete this capability assessment in person. As such, we had to develop a simple protocol to be completed remotely via video call. This protocol tested the participant's ability to reach an approximation of their personalised IE training angle and hold for 60 s. The easy-to-follow instructions allow participants to carry out the test independently. Delivering the test remotely required additional risk assessment, the translation of safety considerations into the home (eg, a nearby chair for support), along with additional online instructional materials.

The IIET stayed the same apart from the time delay of having to establish the status of the exercise type as non-aerosol generating; expert consensus from the Physiological Society was not available until 20 May 2020.[24] However, new PPE considerations had to be implemented immediately in line with government guidelines.[25] This had numerous implications, not least equipment costs and additional time considerations during face-to-face testing.

### Impact of COVID-19 on NHS primary care staff participation

It was originally planned to recruit 2–4 primary care sites in the Southeast based on feasibility searches performed before the pandemic. Between November 2019 and February 2020, one site had committed in principle as a research site. The onset of COVID-19 and unprecedented demands on the NHS, in particular, primary care, led to initial difficulties in identifying principal investigators at prospective sites due to uncertainty of workload. Identification of appropriate HCPs with the capacity to deliver the intervention was already a challenge. This was exacerbated by the fact that GP principal investigators were focused on the COVID-19 response and, later, COVID-19 intervention studies and vaccination.

Embedding clinical research in the NHS is a key theme in the vision of *The Future of UK Clinical Research Delivery*.[14] To create a research-positive culture in which all health and care staff feel empowered to support and participate in clinical research as part of their job role, much greater funding and resourcing of primary care would be necessary. To try and mitigate this in the current study, we were forced to approach sites further afield and would strongly recommend overplanning the number of sites in future exercise-based studies.

### CONCLUSIONS

Delivery of clinical trials in a safe and reliable way has always been complex, requiring good governance and ethical frameworks as well as robust infrastructure. While there are many randomised controlled trials of exercise either published or planned, their use is more limited than conventional trials of medicinal products. In addition, there are barriers to the prescription of exercise by HCPs. These issues became more acute during the COVID-19 pandemic where, quite reasonably, trials directed at intervention in COVID-19 were prioritised.[3] However, it is evident that abandoning preventative healthcare measures has had (and more concerningly will continue to have) a deleterious effect on the general population. In context, maintaining healthy lifestyle is important and this itself could be a protective factor during a pandemic like COVID-19 where patients with obesity and other risk factors were affected more.

We have discussed several predictable hurdles the pandemic created for recruitment to a feasibility study of IE. Other unexpected problems have also arisen, such as a significant reduction in the number of people identified with stage 1 hypertension. Ironically, the pandemic presented opportunities such as unprecedented speed and fluidity of change to the study approach. Remote consent and screening of patients, automated reminders and video validation of BP technique were all developed, approved and tested more rapidly as a result of necessity. However, this impacted directly on our original commitment to ensure equality of access due to the associated IT requirements and level of IT literacy required to engage remotely, for example, need for webcams, two screens open, etc. Overall, a willingness to constantly reflect and respond with protocol changes is essential in the current climate.

Since we were unable to identify eligible patients through primary care, we sought and gained approval for direct marketing of the study resulting in a tremendous response (1362 click-throughs from 63 days of active social media advertising), indicating public willingness and enthusiasm for this type of research. Central databases, opt-in to research and direct marketing (where appropriate), are likely to be much more effective methods for future study recruitment.

Finally, it may be worth considering a consensus statement from leaders in the field of exercise research to find common ways to enhance recruitment to trials of exercise to augment current clinical practice.

In closing, while this study is still ongoing due to the delays caused by COVID-19, it is evident that we would not have been able to achieve our recruitment targets and the necessary data collection without successfully implementing the changes discussed.

**Author affiliations**
[1]Centre for Health Services Studies, University of Kent, Canterbury, UK
[2]Renal Department, East Kent Hospitals University NHS Foundation Trust, Canterbury, UK
[3]Public Co-applicant, Canterbury, UK
[4]Faculty of Health and Wellbeing, Canterbury Christ Church University, Canterbury, UK
[5]Sport, Exercise and Rehabilitation Sciences, Canterbury Christ Church University, Canterbury, UK
[6]Newton Place Surgery, Faversham, UK
[7]Research and Development, Epsom and St Helier University Hospitals NHS Trust, London, UK

**Contributors** CF, JW, MR-R, JO and DM designed the original study concept and design. COVID-19 adaptations were implemented by CF and JW as co-leads

during COVID-19 and learnings collated by ES for this paper. The first draft of the manuscript was written by CF, JW, ES and further developed by MR-R, AW, JD. The paper was then reviewed by DM, VS, KG, TP-H, RB, TD and JO. The study has two lay co-applicants (AW and JD) who are involved as members of the research team and make shared decisions on the study. AW and JD directly contributed to drafting and reviewing the content of this manuscript on Patient and Public Involvement during COVID. PAS is chair of the Study Steering Committee and with TD reviewed drafts of this manuscript for clinical aspects within national context of hypertension treatment.

**Funding** This work was supported by the NIHR research for Patient Benefit (RfPB) Programme (grant number NIHR200485). The views expressed are those of the authors and not necessarily those of the NIHR or the Department of Health and Social Care. Additional funding was also granted by Alan Squirrell Artificial Kidney Unit Trust Charity number 254317. Trial Registration: ISRCTN, ISRCTN13472393. Registered 9 September 2020, https://doi.org/10.1186/ISRCTN13472393.

**Competing interests** PAS has received speaker honorarium from AstraZeneca & Bayer and is a Trustee of the charities Blood Pressure UK and British and Irish Hypertension Society. No conflicts of interest/competing interests were declared by any of the other authors.

**Patient consent for publication** Obtained.

**Ethics approval** This study involves human participants and was approved by Ethics name: National Health Service (NHS) and Health Research Authority (HRA) Ethics Committee. Ethics ID: Research Ethics Committee number 20/L0/0422. Participants gave informed consent to participate in the study before taking part.

**Provenance and peer review** Not commissioned; externally peer reviewed.

**ORCID iDs**
Douglas MacInnes http://orcid.org/0000-0002-3723-7859
Jonathan Wiles http://orcid.org/0000-0002-7790-8063

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
