## [Reviewer comments · BMJ Open]

ARTICLE DETAILS

TITLE (PROVISIONAL)	Delivering clinical studies of exercise in the COVID-19 pandemic: Challenges and adaptations using a feasibility trial of isometric exercise to treat hypertension as an exemplar.
AUTHORS	Farmer, C; Santer, Ellie; West, Alan; Darby, John; Rees-Roberts, Melanie; Doulton, Timothy; MacInnes, Douglas; O'Driscoll, Jamie; Borthwick, Rachel; Pellatt-Higgins, T; Gousia, Katerina; Short, Vanessa; Swift, Pauline; Wiles, Jonathan

VERSION 1 – REVIEW

REVIEWER	Meresh, Edwin Loyola University Medical Center
REVIEW RETURNED	06-Jan-2023

GENERAL COMMENTS	Very important topic related to the pandemic. Page 4, line 88, please add obesity to the risk factors. Page 11, line 244, please consider adding the following: Maintaining healthy life style is important and this itself could be a protective factor during a pandemic like COVID-19 were patients with obesity and other risks factors were affected more.
---

REVIEWER	Gulliford, Martin King's College London, UK
REVIEW RETURNED	24-Jan-2023

GENERAL COMMENTS	This is a thoughtful and well written article that provides a critical reflection on the experience of delivering a physical activity trial during the Covid-19 pandemic. The article provides a useful resource to other researchers and research managers. The end of the abstract and paper could explain whether the trial achieved a successful outcome in terms of completion. • Are the issues raised by the article important to BMJ Open's broad and international readership that includes patients, researchers, policy makers, health professionals, and doctors of all disciplines? Yes• Is the article interesting and offering novel insights that have not been sufficiently considered in the existing published literature? Yes
---

	 • Is the article well written and is the content clearly presented? Does it have a clear message? Yes  • Will the article help medical researchers, patients or related groups of readers to make better decisions? Yes, in all probability.  • Does the article demonstrate one or more of the following values: transparency, openness, collaboration, innovation, reproducibility, patient/ public involvement, improving peer review and journal best practice, and reducing research waste? Yes
--	---

VERSION 1 – AUTHOR RESPONSE

Reviewer: 1

Dr. Edwin Meresh, Loyola University Medical Center

Comments to the Author:

Very important topic related to the pandemic. Page 4, line 88, please add obesity to the risk factors. Obesity has now been added as another example of a cardiovascular disease risk factor.

Page 11, line 244, please consider adding the following: Maintaining healthy life style is important and this itself could be a protective factor during a pandemic like COVID-19 were patients with obesity and other risks factors were affected more.

This sentence has now been added at the end of the paragraph as requested.

Reviewer: 2

Dr. Martin Gulliford, King's College London, UK

Comments to the Author:

This is a thoughtful and well written article that provides a critical reflection on the experience of delivering a physical activity trial during the Covid-19 pandemic. The article provides a useful resource to other researchers and research managers. The end of the abstract and paper could explain whether the trial achieved a successful outcome in terms of completion.

Thank you for your confirmation that the article provides a useful resource to other researchers and research managers, we have been able to use this view as a strength of the paper.

The final sentence of the abstract now reads:

In conclusion, it is essential to reflect and respond with protocol changes in the current climate in order to deliver clinical research successfully; as in the case of this particular study.

The final sentence of the conclusion now reads:

In closing, whilst this study is still ongoing due to the delays caused by COVID-19, it is evident that we would not have been able to achieve our recruitment targets and the necessary data collection without successfully implementing the changes discussed.

- Are the issues raised by the article important to BMJ Open's broad and international readership that includes patients, researchers, policy makers, health professionals, and doctors of all disciplines?

Yes

- Is the article interesting and offering novel insights that have not been sufficiently considered in the existing published literature?

Yes

- Is the article well written and is the content clearly presented? Does it have a clear message?

Yes

- Will the article help medical researchers, patients or related groups of readers to make better decisions?

Yes, in all probability.

- Does the article demonstrate one or more of the following values: transparency, openness, collaboration, innovation, reproducibility, patient/ public involvement, improving peer review and journal best practice, and reducing research waste?

Yes